# The Use of DArTseq Technology to Identify New SNP and SilicoDArT Markers Related to the Yield-Related Traits Components in Maize

**DOI:** 10.3390/genes13050848

**Published:** 2022-05-10

**Authors:** Agnieszka Tomkowiak, Bartosz Nowak, Aleksandra Sobiech, Jan Bocianowski, Łukasz Wolko, Julia Spychała

**Affiliations:** 1Department of Genetics and Plant Breeding, Poznan University of Life Sciences, Dojazd 11, 60-632 Poznan, Poland; agnieszka.tomkowiak@up.poznan.pl (A.T.); julia.spychala@up.poznan.pl (J.S.); 2Poznan Plant Breeding Sp. z o. o., Kasztanowa 5, 63-004 Tulce, Poland; nowak@hrsmolice.pl; 3Department of Mathematical and Statistical Methods, Poznan University of Life Sciences, Wojska Polskiego 28, 60-637 Poznan, Poland; jan.bocianowski@up.poznan.pl; 4Department of Biochemistry and Biotechnology, Poznan University of Life Sciences, Dojazd 11, 60-632 Poznan, Poland; lukwolko@up.poznan.pl

**Keywords:** maize, yield, molecular markers, NGS

## Abstract

In the last decade, many scientists have used molecular biology methods in their research to locate the grain-yield-determining loci and yield structure characteristics in maize. Large-scale molecular analyses in maize do not only focus on the identification of new markers and quantitative trait locus (QTL) regions. DNA analysis in the selection of parental components for heterotic crosses is a very important tool for breeders. The aim of this research was to identify and select new markers for maize (SNP and SilicoDArT) linked to genes influencing the size of the yield components in maize. The plant material used for the research was 186 inbred maize lines. The field experiment was established in twolocations. The yield and six yield components were analyzed. For identification of SNP and SilicoDArT markers related to the yield and yield components, next-generation sequencing was used. As a result of the biometric measurements analysis, differentiation in the average elevation of the analyzed traits for the lines in both locations was found. The above-mentioned results indicate the existence of genotype–environment interactions. The analysis of variance for the observed quality between genotypes indicated a statistically significant differentiation between genotypes and a statistically significant differentiation for all the observed properties betweenlocations. A canonical variable analysis was applied to present a multi-trait assessment of the similarity of the tested maize genotypes in a lower number of dimensions with the lowest possible loss of information. No grouping of lines due to the analyzed was observed. As a result of next-generation sequencing, the molecular markers SilicoDArT (53,031) and SNP (28,571) were obtained. The genetic distance between the analyzed lines was estimated on the basis of these markers. Out of 81,602 identified SilicoDArT and SNP markers, 15,409 (1559 SilicoDArT and 13,850 SNPs) significantly related to the analyzed yield components were selected as a result of association mapping. The greatest numbers of molecular markers were associated with cob length (1203), cob diameter (1759), core length (1201) and core diameter (2326). From 15,409 markers significantly related to the analyzed traits of the yield components, 18 DArT markers were selected, which were significant for the same four traits (cob length, cob diameter, core length, core diameter) in both Kobierzyce and Smolice. These markers were used for physical mapping. As a result of the analyses, it was found that 6 out of 18 (1818; 14,506; 2317; 3233; 11,657; 12,812) identified markers are located inside genes. These markers are located on chromosomes 8, 9, 7, 3, 5, and 1, respectively.

## 1. Introduction

In the last century, scientific and technological progress has contributed significantly to the growth of the global food supply. United Nations (UNO) predictions indicate that at the current rate of population growth, the number of people in the world will exceed 9 billion in 2050 [1]. At the same time, the amount of arable land will decrease, so the demand for high-yield varieties will increase. This will include maize, which, along with rice and wheat, provides at least 30% of the calories in food for over 4.5 billion people in 94 developing countries. In some parts of Africa and America, corn only provides more than 20% of the calories in food. This grain is also a crucial ingredient in animal feed and is widely used in industrial products, including biofuel production [2,3].

As a cross-pollinated crop, genomic divergence is nearly 1.42% between two inbred maize lines, which is greater than the divergence of 1.34% between humans and chimpanzees [4]. The large genetic diversity of maize makes this plant a very good material for genetic improvement. The price recession in sequencing and the rapid development of next-generation sequencing technology have opened a new research opportunity in the field of functional maize genomics, particularly since the publication of the maize B73 reference genome in 2009, the draft genome of the elite inbred maize line “Ph207” has recently been developed through Illumina Tru-seq Synthetic Long-Read technology [5,6]. Two new maize genomes were assembled and released by 2018 [7,8]. Now, Maize GDB contains 12 complete maize genomes and is expected to increase to 40. Due to the dynamic progress in maize breeding, many functional genes influencing, inter alia, agronomic traits, were cloned and used in breeding programs [9]. Previously, an increase in yields was obtained through the use of higher doses of fertilizer, which, of course, was not indifferent to the environment [10,11]. Currently, the goal is to increase the yield of plants by using biotechnological tools in plant breeding.

Many reference genomes for crop plants have been generated over the past decade, but these genomes are often fragmented and missing complex repeat regions. In their research, Jiao et al. [12] showed that characterization of the repetitive portion of the genome revealed more than 130,000 intact transposable elements, allowing them to identify transposable element lineage expansions that are unique to maize.

Thanks to next-generation sequencing technology, we can generate huge amounts of DNA sequence data, which will become a tool for the identification of high-density molecular markers. Then, these markers can be used in breeding programs to select lines that yield well. Techniques such as next-generation sequencing (NGS), quantitative trait locus (QTL) mapping, genome-wide association studies (GWAS), and nested association mapping (NAM) are routinely used to dissect complex traits such as yield, plant size or architecture, pathogen resistance, and control of metabolic pathways [13,14,15,16].

Using next-generation sequencing in plants under drought stress, more than 100 SNP markers for root traits were identified [17,18]. Similar studies on drought stress were carried out [19] where they identified the QTL and SNP of drought-related traits. Muraya et al. [20] identified a total of 383,145 SNPs associated with maize biomass in 21 inbred lines. Using NGS and association mapping, 261,055 SNPs related to the tar leaf spot in maize were identified [21].

The aim of this research was to identify and select new markers for maize selection (SNP and SilicoDArT) linked to genes influencing the size of the yield components in maize.

## 2. Results

### 2.1. Analysis of the Size of the Components of the Yield of Inbred Maize Lines

Setting-up of the field experiment made it possible to perform and analyze the biometric measurements of 186 inbred lines. In the next stage of analyzes, these measurements were used for association mapping. After the harvest, the following yield components were observed: cob length (CL), cob diameter (CD), core length (COL), core diameter (COD), the number of rows of grain (NGR), the number of grains in a row, weight of one thousand grains (WTG), and yield. The yield from each plot was also analyzed. All the observed traits had normal distribution. Analysis of variance indicated that the main effects of line and location as well as line × location interaction were significant for all the traits of study. Three traits (cob diameter, core length and the number of rows of grain) were characterized by greater variability in Kobierzyce than in Smolice (Appendix A). Table 1 shows the means for all lines simultaneously for each feature in bothlocations. We observe diversity in the average height of the analyzed components, e.g., the average cob length for all lines in Smolice was 13.1 cm and the average cob length for the same lines in Kobierzyce was 15.4 cm. It was similar in the case of other components, where the average mass of grain from the in Smolice was 91.2 g, and in Kobierzyce—111 g (Table 1). The above results indicate the existence of genotype-environmental interactions.

#### 2.1.1. Multi-Traits Comparisons

To determine the relationships between lines on the basis of all observed traits, a multivariate technique was used, that is, canonical variable analysis (Figure 1). The values for the first two canonical variables were significant and jointly accounted for 62.16% of the whole variation (Figure 1). The greatest variation in terms of all traits jointly measured with Mahalanobis distances was found for lines 59 and 71 (the distance between them amounted to 15.366). The greatest similarity was found between 31 and 122 (0.520) (Table 2).

No grouping of lines was observed due to the analyzed characteristics and origin from a given breeding company. Both canonical variables were significantly discriminant for CL, CD, COL, WTG, COD, mass of grain from the cob, NRG, and yield.

#### 2.1.2. Relationships between Traits

The positive, statistically significant correlations in both locations were observed between 25 pairs of traits (Figure 2 and Figure 3). Negative correlations in both locations were observed between two pairs of traits (Figure 2 and Figure 3). Additionally, the number of grains in a row was positively correlated with the number of rows of grain only in Kobierzyce (Figure 2); however, core length was negatively correlated with the number of rows of grain only in Smolice (Figure 3).

### 2.2. Next-Generation Sequencing to Identify SNP and SilicoDArT Markers Related to Maize Yield Genes

A total of 186 lines were sent for next-generation sequencing and were also analyzed in the field. As a result of sequencing, the molecular markers SilicoDArT (53,031) and SNP (28,571) were obtained, on the basis of which the genetic similarity between the analyzed inbred corn lines was estimated (Figure 4). Based on genetic similarity, the analyzed lines formed four main groups.

In group I, there are two lines (K037 and K038) from Plant Breeding Kobierzyce, which were 63% similar to each other. These lines were 41% similar to the remaining 184 genotypes (Figure 4). The second group includes two lines (S145 and S132) belonging to Plant Breeding Smolice. These lines are 51% similar to each other and 45% similar to the other lines (Figure 4). The third group consists of 73 lines. Within the third group, three essential subgroups can be distinguished. The first subgroup consists of 25 lines (2 belong to Plant Breeding Kobierzyce and 23 to Plant Breeding Smolice). The second subgroup consists of 23 lines (2 belong to Plant Breeding Kobierzyce and 21 to Plant Breeding Smolice). The first and second subgroups are 58% similar. The third subgroup is 49% similar to the first and second and includes 25 lines (5 belonging to Plant Breeding Kobierzyce and 20 belonging to Plant Breeding Smolice) (Figure 4). The fourth group is also made up of three subgroups (109 lines in total). The first subgroup consists of 23 lines (14 belonging to Plant Breeding Kobierzyce and 9 belonging to Plant Breeding Smolice). The lines from HR Kobierzyce are 59% similar to each other and 49% similar to the lines from HR Smolice. The second subgroup consists of 26 lines (11 from Plant Breeding Kobierzyce and 15 from Plant Breeding Smolice). The lines from Plant Breeding Smolice are 60% similar to each other, while they are 51% similar to the lines from Plant Breeding Kobierzyce (Figure 3). The third most numerous subgroup includes 59 lines from Plant Breeding Smolice and 1 from Plant Breeding Kobierzyce (K008). The lines from Plant Breeding Smolice are from 59% to 95% similar to each other (Figure 4). Analyzing the dendrogram, it can be noticed that the lines from Smolice show greater similarity with each other than with the lines from Kobierzyce, and conversely, the lines from Kobierzyce are more similar to each other than to the lines from Smolice.

### 2.3. Association Mapping Using GWAS Analysis

As a result of next-generation sequencing, a total of 81,602 molecular markers (53,031 SilicoDArT and 28,571 SNP) were obtained, of which 15,409 (1559 SilicoDArT and 13,850 SNP) were selected as a result of association mapping, which showed them to be significantly related to the analyzed traits of the yield structure itself (Table 3). The greatest number of molecular markers was associated with cob length (1203), cob diameter (1759), core length (1201), and core diameter (2326). The fewest markers were associated with the number of rows of grain (321) and the number of grains in a row (130) (Table 3). In order to narrow down the number of markers for physical mapping, among all the significant ones, 16 were selected, which were related to the same four traits in both locations (Kobierzyce and Smolice).

### 2.4. Physical Mapping and Functional Analysis of Gene Sequences

From 15,409 (1559 SilicoDArT and 13,850 SNP) markers significantly related to the analyzed yield components, 18 were selected that were significant for the same four traits in both locations (Kobierzyce and Smolice) (Table 4). An attempt was also made to determine the location of the selected DArT markers. As a result of the analyses, it was found that 6 out of 18 (1818; 14,506; 2317; 3233; 11,657; 12,812) of the selected markers are located inside the genes, as described in Table 4; for the remaining 12 markers, their location and distance from the nearest located genes are shown.

## 3. Discussion

Since the mid-1990s, intensive research has been conducted in many centers around the world in terms of the structure and function of the maize genome, using modern biotechnology and molecular biology methods. As a result of comprehensive breeding experiments, phenotypic observations, and genetic analyses, the QTL associated with specific quantitative traits were identified. Along with advancements in the development of high-efficiency DNA sequencing methods, enabling the sequence of whole genomes and transcriptomes to be known, a new quality of research has emerged in many plant species, including maize [22,23].

The introduction of NGS methods made it possible to discover the nucleotide se-quence of plants other than model organisms with a small genome, such as *Arabidopsis thaliana*. The main areas of interest are cultivated species such as cereals, coffee, maize, and sugar cane [24]. Since the sequencing of the genome of the first model plant in 2000, the sequences of more than 100 other plant species have been recorded [25,26]. Thanks to these tests, it is possible to detect SNP polymorphisms and their correlation with specific trait features, as well as the so-called genomic selection, which enables the monitoring of entire genome segments in recombinant breeding programs. It is important that the practical goals imply undertaking comprehensive basic research, making a significant contribution to the development of knowledge in the field of genetics, physiology, and biochemistry of plants [27].

The authors of this work, as a result of next-generation sequencing, obtained the molecular markers SilicoDArT (53,031) and SNP (28,571), on the basis of which the genetic distance between the analyzed inbred maize lines was estimated. Based on genetic similarity, the analyzed lines formed four main groups. When analyzing the dendrogram, it can be noticed that the lines from Plant Breeding Smolice show greater similarity with each other than with the lines from Plant Breeding Kobierzyce, and vice versa, the lines from Kobierzyce are more similar to each other than to the lines from Smolice.

Another sequencing strategy, mainly used to study plant–environment interactions, is to use NGS methods to understand the plant transcriptome under specific physiological states. By analyzing the cDNA sequences, one obtains information about the sequence expression tags (ESTs) that are transcribed in particular tissues and organs; despite some limitations, these data are very useful for breeders [28].

The authors of this study chose 15,409 (1559 SilicoDArT and 13,850 SNP) from among 81,602 identified SilicoDArT and SNP markers significantly related to the analyzed yield components itself. The greatest number of molecular markers was associated with cob length (1203), cob diameter (1759), core length (1201), and core diameter (2326).

SNP and SilicoDArT markers have many other applications, including creating molecular maps of couplings or finding QTL (quantitative trait locus) areas that are responsible for the inheritance of quantitative yield components. In addition, they are used for origin analysis, searching for the “fingerprint” of breeding varieties, in studies of genetic diversity in populations, gene flow, and plant evolutionary genetics [29].

The conducted research allowed the authors of this publication to choose from 15,409 markers significantly related to the analyzed trait of the yield structure of 16 DArT markers, which were significant for the same four traits in both locations for Kobierzyce and Smolice. These markers were used for physical mapping. As a result of the analyses, it was found that 6 out of 16 (1818; 14,506; 2317; 3233; 11,657; 12,812) selected markers are located inside genes. These markers are located on chromosomes 8, 9, 7, 3, 5, and 1, respectively. For the remaining markers, their location and distance from the closest genes are given.

Marker 1818, which was significantly related to four yield components, is located inside the *cinnamoyl-CoA reductase* gene. *Cinnamoyl-CoA reductase* (CCR) is considered a key enzyme in controlling the quantity and quality of lignins. The first transgenic plants with reduced CCR activity were obtained in tobacco. The lignin content was reduced by 50% compared to the wild type. The decrease in lignin had a detrimental effect on the development of the tobacco plants. CCR biosynthesis has been extensively characterized in dicots, but not in monocots [30]. In *Arabidopsis thaliana*, two recombinant proteins are responsible for three cinnamoyl-CoA reactions, but with different levels of efficiency [31].

The identified significant marker 14,506 is located inside the gene (WAT1-related protein At1g09380). Based on transcriptomic, metabolomic, and physiological data, WAT1 has been found to be involved in the integration of auxin signaling and secondary cell wall formation in *Arabidopsis* [32].

Another significant marker, 2317, is also located inside the gene (*eukaryotic translation initiation factor 3 subunit c*). Translation regulation mainly focuses on the initiation phase. There, one of the important initiation factors is the large protein complex of the eIF3. In all eukaryotes, the general function of eIF3 is to form a translation initiation complex scaffold and to increase the accuracy of the scanning mechanism for selection of start codons. Over the past decades, additional functions of eIF3 have been described as necessary for the development of various eukaryotic organisms, including plants. The plant architecture of the eIF3 complex is similar to that of the mammalian eIF3. Several plant eIF3 subunits have been analyzed over the past 20 years, mainly using *Arabidopsis* as a model organism [33].

Marker 3233, which was significantly associated with four yield components, is also located inside the gene (*RNA polymerase II transcriptional coactivator KELP*). In *Arabidopsis thaliana* AtKELP, it has the ability to bind to tomato mosaic virus (ToMV) and may interfere with the movement of the virus to the cell. From the virus protection point of view, it is anticipated that KELP and its inhibitory mechanisms can be used to generate new plant resistance to viruses [34].

The identified marker 11,657 is anchored inside the *aspartate aminotransferase* gene. During research on rice, wheat, and barley, it was found that the action of *aspartate aminotransferase* increases the yield. The authors also showed that the asparagine content in the roots and shoots of mutant rice plants was reduced compared to wild-type plants [35].

The marker 12,812 is anchored inside the gene (*sucrose transporter 1*). From genomic data, nine sucrose transport proteins have been identified in *Arabidopsis*, seven in maize, five in poplar, and five in rice. These genes encoding sucrose transporters are usually called SUT or SUC. Currently, only the entire SUT or SUC *Arabidopsis* genes have been extensively studied. AtSUC1 is highly concentrated in pollen and is important for sucrose-dependent signaling, which may lead to anthocyanin accumulation in seedlings and the normal function of male gametophytes [36].

For several years now, maiz breeding worldwide has been aided by useful molecular markers. Many authors state in their publications that molecular-marker-assisted breeding is accelerating yield gains not only in the USA but also in other countries, offering tremendous potential for enhancing the productivity and value of maize germplasm [37,38]. Authors discuss the importance of efforts in meeting the growing demand for maize and provide examples of the recent use of molecular markers with respect to DNA fingerprinting and genetic diversity analysis of maize germplasm, QTL analysis of important biotic and abiotic stresses, and marker-assisted selection (MAS) for maize improvement [39,40,41,42]. The above-mentioned markers, especially 11,657 related to aspartate aminotransferase, which may increase the yield, can be used for the selection of successful maize lines. The modern biotechnological tools for genomic (i.e., genome editing) could accelerate maize breeding, thus contributing to global food security.

## 4. Materials and Methods

### 4.1. Plant Material

Plant material consisted of 186 inbred lines derived from Plant Breeding Smolice Sp. z o.o. IHAR Group and Małopolska Plant Breeding Sp. z o.o. These lines (186) were deployed in two locations, 120 km apart: Smolice, 51°41′23.16″ N 17°4′ 18.241″ E; and Kobierzyce, 50°58′19,411″ N 16°55′47,323″ E.

### 4.2. Methods

#### 4.2.1. Phenotyping

A field experiment with 186 inbred lines was set up on plots of 10 m^2^, in a system of complete randomly selected blocks, for three repetitions, in twolocations. During the experiments, observations of morphological traits were carried out, and after the harvest, in the first half of November, biometric measurements were made, which included the following traits: cob length, cob diameter, core length, core diameter, number of rows of grain, number of grains in a row, weight of one thousand grains (WTG), and yield. Measurements were carried out on ten randomly selected cobs from three replicates.

#### 4.2.2. DNA Isolation

DNA isolation from 186 inbred lines was made by using Wizard^®^ Genomic DNA Purification Kit from the Promega company. The concentration and purity of the isolated DNA samples were determined using a DS-11 spectrophotometer from the DeNovix company. The isolated template DNA was adjusted to an equal concentration of 100 ng/μL by dilution with distilled water.

#### 4.2.3. Genotyping

The DArTseq analysis was performed at Diversity Arrays Technology Pty Ltd. (Australia). The methodology presented below was also used in the research presented by Tomkowiak et al. [43].

DNA sample digestion/ligation reactions were processed according to Kilian et al. [44], but replacing a single PstI-compatible adaptor with two adaptors corresponding to PstI- and NspI-compatible sequences and moving the assay on the sequencing platform as described by Sansaloni et al. [45]. The PstI-compatible adapter was designed to include Illumina flowcell attachment sequence, sequencing primer sequence, and “staggered” varying length barcode region, similar to the sequence reported by Elshire et al. [46]. Reverse adapter contained flowcell attachment region and NspI-compatible overhang sequence. Only “mixed fragments” (PstI–NspI) were amplified in PCR using the following reaction conditions: denaturation, 1 min at 94 °C; followed by 30 cycles of 94 °C for 20 s, 57 °C for 30 s, and 72 °C for 45 s; and the final elongation, 72 °C for 7 min. After PCR, equimolar amounts of amplification products from each sample of the 96-well microtiter plate were bulked and applied to c-Bot (Illumina) bridge PCR, followed by sequencing on Illumina Hiseq2500. The sequencing (single read) was run for 78 cycles. Sequences generated from each lane were processed using proprietary DArT analytical pipelines. In the primary pipeline, the fastq files were first processed to filter away poor-quality sequences, applying more stringent selection criteria to the barcode region compared to the rest of the sequence. In that way, the assignments of the sequences to specific samples carried in the “barcode split” step were very reliable. Approximately 2,500,000 (±7%) sequences per barcode/sample were used in marker calling. Finally, identical sequences were collapsed into “fastqcall files”. These files were used in the secondary pipeline for DArT PL’s proprietary SNP and SilicoDArT (presence/absence of restriction fragments in representation) calling algorithms (DArTsoft14). For the association analysis, only DArT sequences meeting the following criteria were selected: one SilicoDArT and SNP within a given sequence (69 nt), minor allele frequency (MAF) > 0.25, and missing observation fractions < 10%.

#### 4.2.4. Statistical Analysis and Association Mapping Using GWAS Analysis

The normality of the distribution of the observed traits was tested using Shapiro–Wilk’s normality test to check whether the analysis of variance (ANOVA) met the assumption that the ANOVA model residuals followed a normal distribution. The homogeneity of variance was tested using Bartlett’s test. A two-way analysis of variance (ANOVA) was carried out to determine the main effects of line, location, and line-by-location interaction on the variability of the studied traits. The genetic similarity for each pair of the investigated lines was estimated based on the coefficient proposed by Nei and Li [47]. The lines were grouped hierarchically using the unweighted pair group method of arithmetic means (UPGMA) based on the calculated coefficients [48]. The relationships between the lines were presented in the form of a dendrogram. The relationships between observed traits were assessed based on Pearson’s correlation coefficients and tested with the *t*-test, independently for experiments in Kobierzyce and Smolice. The results were also analyzed using multivariate methods. A canonical variable analysis (CVA) was applied to present a multi-trait assessment of the similarity of the tested lines in a lower number of dimensions with the lowest possible loss of information. Mahalanobis distance was suggested as a measure of “polytrait” mazie lines similarity [49], the significance of which was verified by means of critical value Dα, also called “the least significant distance”. Mahalanobis distances were calculated for all lines.

By means of GWAS analysis, an association mapping was made for the yield structure characteristics of 186 maize genotypes. This mapping was performed on the basis of the results obtained from genotyping and phenotyping. The genotypic data were obtained from the DArTseq analysis, while the phenotypic data were the field results concerning the yield components. The analyzed yield components were cob length, cob diameter, core length, core diameter, number of rows of grain, number of grains in a row, weight of one thousand grains, and yield. The number of significant markers associated with each particular trait, range of significant effects (minimal and maximal values), mean value of all significant effects, and sum of significant effects were presented in Table 3. Based on the GWAS analysis, silicoDArT and SNP markers with the highest significance level were selected for further research, that is, those that were most strongly associated with the yield structure. All analyses were conducted in Genstat 18.2 (VSN International Ltd., Hemel Hempstead, England, UK).

#### 4.2.5. Association Mapping

The sequences of the silicoDArT and SNP markers selected on the basis of the GWAS analysis were subjected to BLAST (Basic Local Alignment Search Tool). This consisted of searching databases for sequences with high homology to the selected sequences of the silicoDArT and SNP markers. These analyses were performed on the URGI (Unité de Recherche Génomique Info) website with a completely sequenced maize genome. The URGI program was used to indicate the chromosomal locations of the searched sequences similar to the analyzed sequences and determine their physical location. In order to identify the most likely region containing the sequences most similar to the analyzed sequences, an overall probability was calculated from the e-value of each chromosome. The sequences of all genes in the designated area on the chromosome were subsequently analyzed.

#### 4.2.6. Functional Analysis of Gene Sequences

Functional analysis was made in the Blast2GO program. The sequences of all genes located in the area of chromosomes determined on the basis of the BLAST analysis performed on the URGI website were analyzed. The aim was to obtain information on the biological function of the sequence of genes located in the designated region of the chromosomes.

## 5. Conclusions

As a result of the analysis of biometric measurements, differentiation was found in the mean height of the analyzed traits for the lines in Smolice and Kobierzyce. The above results indicate the existence of genotype–environment interactions. The analysis of variance for the observed traits between genotypes showed a statistically significant differentiation between genotypes and a statistically significant differentiation for all the observed traits between the locations where the field experiment was carried out (Kobierzyce and Smolice). A canonical variable analysis was applied to present a multi-trait assessment of the similarity of the tested maize genotypes in a lower number of dimensions with the lowest possible loss of information. No grouping of lines due to the analyzed traits was observed. The analysis of the correlation of the examined traits showed that in both Smolice and Kobierzyce, the mass of grain from the cob and the yield per plot (100%), as well as the length cob and the length core, were positively correlated the most positively. As a result of next-generation sequencing, the molecular markers SilicoDArT (53,031) and SNP (28,571) were obtained, on the basis of which the genetic distance between the analyzed lines was estimated. Based on genetic similarity, the analyzed lines formed four main groups. When analyzing the dendrogram, it can be noticed that the lines from Smolice show greater similarity with each other than with the lines from Kobierzyce, and conversely, the lines from Kobierzyce are more similar to each other than the lines from Smolice. Out of 81,602 identified SilicoDArT and SNP markers, 15,409 (1559 SilicoDArT and 13,850 SNPs) significantly related to the analyzed yield components itself were selected as a result of association mapping. The greatest number of molecular markers was associated with cob length (1203), cob diameter (1759), core length (1201), and core diameter (2326). From 15,409 markers significantly related to the analyzed location yield components, 18 DArT markers were selected, which were significant for the same four traits in both Kobierzyce and Smolice. These markers were used for physical mapping. As a result of the analyses, it was found that 6 out of 18 (1818; 14,506; 2317; 3233; 11,657; 12,812) identified markers are located inside genes. These markers are located on chromosomes 8, 9, 7, 3, 5, and 1. In the case of the remaining markers, their location and distance from the closest genes, which are partially characterized, were given. The detailed role of individual proteins will be determined in the next year of research after careful analysis of literature reports.

## Figures and Tables

**Figure 1 genes-13-00848-f001:**
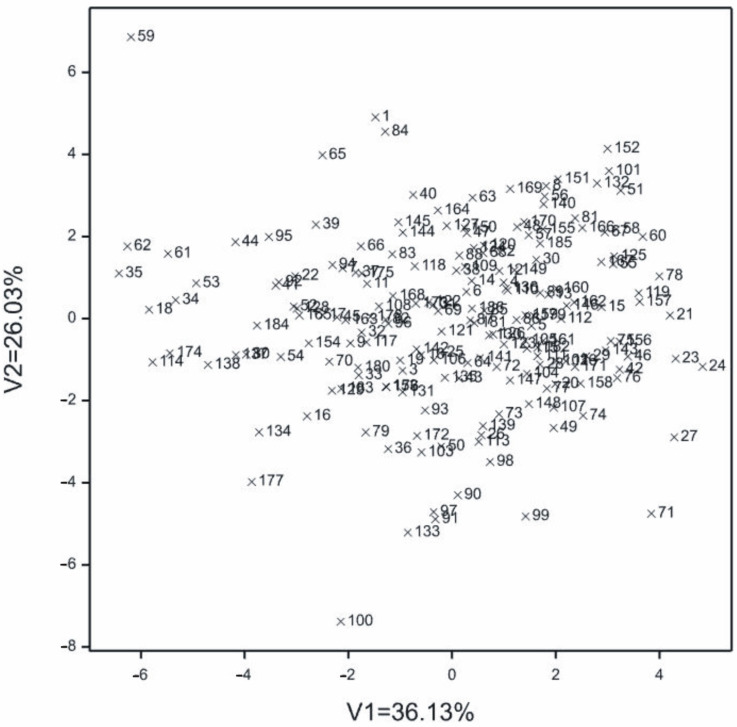
Distribution of individual lines in the space of first two canonical variables on the basis of all observed traits. V1—first canonical variable; V2—second canonical variable.

**Figure 2 genes-13-00848-f002:**
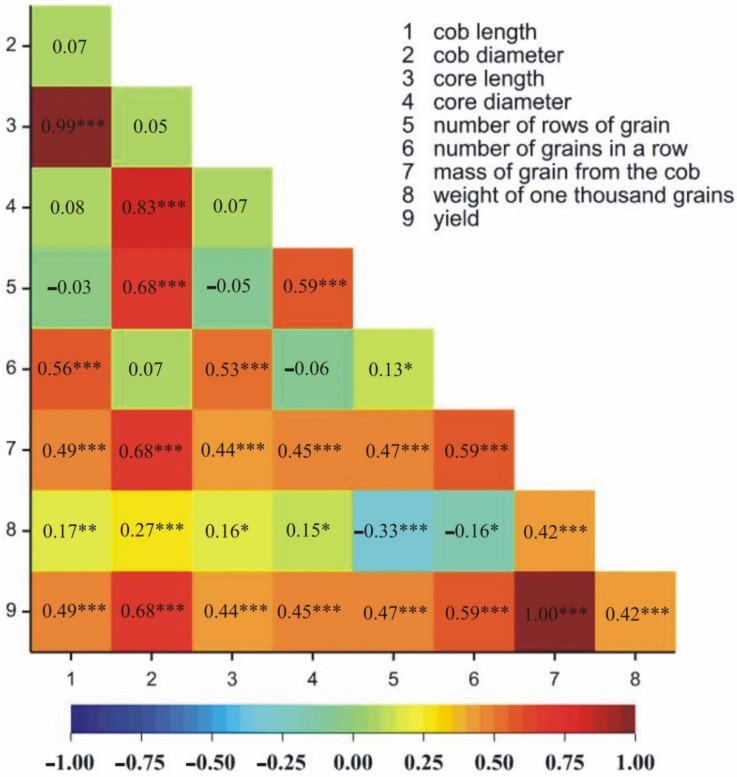
Heatmaps for linear Pearson’s correlation coefficients between the observed components on the basis of mean values for lines in Kobierzyce. * *p* < 0.05, ** *p* < 0.01, *** *p* < 0.001.

**Figure 3 genes-13-00848-f003:**
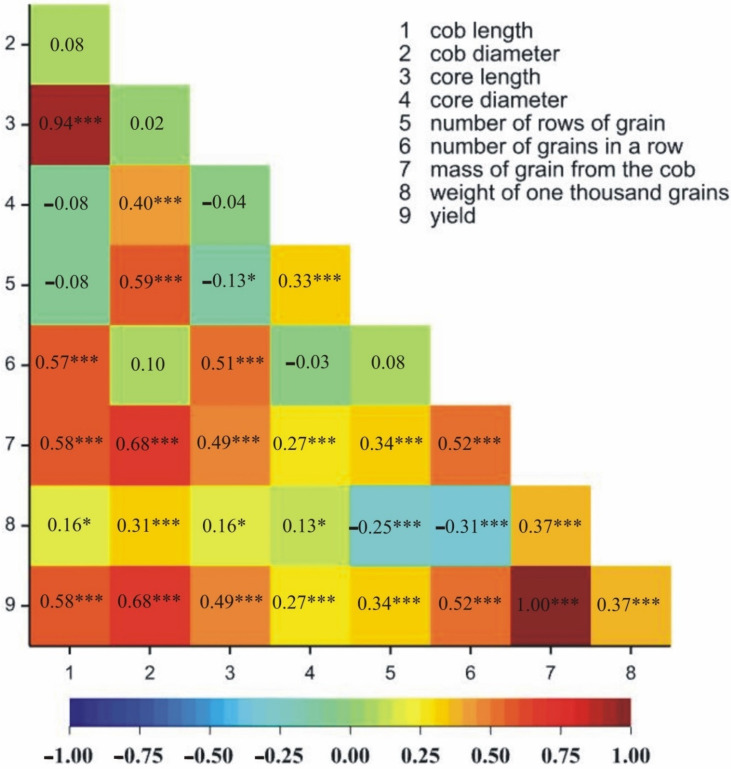
Heatmaps for linear Pearson’s correlation coefficients between the observed components on the basis of mean values for lines in Smolice. * *p* < 0.05, *** *p* < 0.001.

**Figure 4 genes-13-00848-f004:**
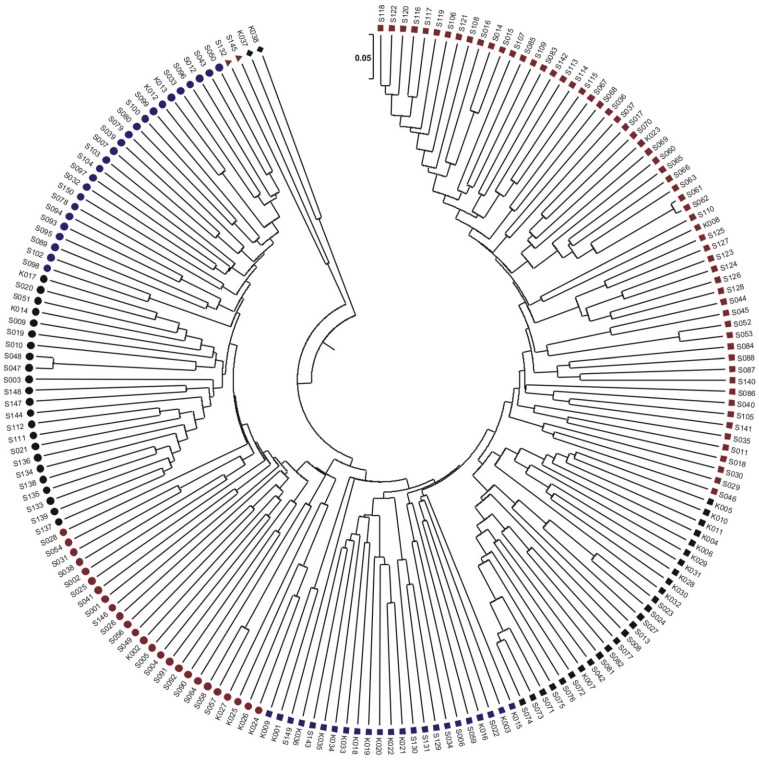
Dendrogram of genetic similarity between the analyzed lines, determined based on all identified molecular markers of SilicoDArT and SNP.

**Table 1 genes-13-00848-t001:** Mean values and standard deviations for all lines for observed traits in Kobierzyce and Smolice.

Location	Smolice		Kobierzyce	
Trait	Mean	Standard Deviation	Mean	Standard Deviation
cob length (cm)	13.1	1.5	15.4	1.7
cob diameter (cm)	4.0	0.3	3.9	0.4
core length (cm)	13.4	1.6	15.3	1.8
core diameter (cm)	2.2	0.4	2.1	0.3
the number of rows of grain	14.9	1.9	15.5	2.1
the number of grains in a row	25.0	4.5	27.6	3.4
mass of grain from the cob (g)	91.2	19.1	111.0	20.8
weight of one thousand grains (g)	284.0	46.7	260.0	41.5
yield (kg)	3.6	0.8	4.4	0.8

**Table 2 genes-13-00848-t002:** Minimum and maximum Mahalanobis distances between studied lines calculated on the basis of nine quantitative traits.

Lines Number		Minimal Mahalanobis Distances	Lines Number		Maximal Mahalanobis Distances
31	122	0.520	59	71	15.366
89	160	0.712	59	100	14.982
4	110	0.808	27	59	14.474
109	124	0.889	59	99	13.960
108	178	0.981	24	59	13.884
55	167	1.099	59	133	13.560
38	109	1.103	59	91	13.441
5	179	1.113	59	74	13.263
5	161	1.145	23	59	13.243
101	132	1.186	21	59	13.183
69	96	1.213	59	76	13.163
104	148	1.227	59	90	13.115
38	124	1.245	59	97	13.062
161	179	1.250	100	152	12.965
96	163	1.284	1	100	12.741
158	171	1.291	59	73	12.718
159	162	1.299	42	59	12.656

**Table 3 genes-13-00848-t003:** Molecular markers of SilicoDArT and SNP significantly (LOD > 3.0) related to the analyzed characteristics of the yield structure (significant associations selected at *p* < 0.05 with correction for multiple testing using the Benjamini–Hochberg method). Table contains the number of significant markers associated with each particular trait, range of significant effects (minimal and maximal effects), mean value of all significant effects, and sum of significant effects (total effect).

Location	Trait	The Number of Significant Markers			Minimal Effect			Maximum Effect			Mean Effect			Total Effect		
		DArT	SNP	Total	DArT	SNP	Total	DArT	SNP	Total	DArT	SNP	Total	DArT	SNP	Total
Kobierzyce	cob length	136	1067	1203	−1.234	−1.469	−1.469	1.381	1.574	1.574	0.092	−0.043	−0.028	12.483	−46.064	−33.581
	cob diameter	185	1574	1759	−0.279	−0.305	−0.305	0.269	0.311	0.311	0.039	0.008	0.011	7.146	12.990	20.136
	core length	132	1069	1201	−1.270	−1.497	−1.497	1.404	1.586	1.586	0.117	−0.026	−0.010	15.389	−27.567	−12.178
	core diameter	236	2090	2326	−0.215	−0.227	−0.227	0.225	0.247	0.247	0.043	0.027	0.029	10.225	57.202	67.427
	the number of rows of grain	29	292	321	−1.194	−1.358	−1.358	1.256	1.526	1.526	0.047	−0.046	−0.038	1.374	−13.482	−12.108
	the number of grains in a row	8	122	130	−1.727	−2.145	−2.145	1.727	2.276	2.276	−0.413	0.029	0.001	−3.301	3.489	0.188
	mass of grain from the cob	36	425	461	−14.200	−14.030	−14.200	13.270	15.910	15.910	4.739	0.608	0.931	170.590	258.440	429.030
	weight of one thousand grains	56	462	518	−28.630	−30.290	−30.290	29.920	27.620	29.920	2.466	0.904	1.073	138.110	417.540	555.650
	yield	37	424	461	−0.568	−0.561	−0.568	0.531	0.637	0.637	0.195	0.023	0.037	7.227	9.941	17.168
Smolice	cob length	140	1158	1298	−1.020	−1.265	−1.265	1.254	1.297	1.297	0.050	−0.056	−0.044	6.970	−64.653	−57.683
	cob diameter	147	1411	1558	−0.240	−0.263	−0.263	0.213	0.237	0.237	0.008	−0.004	−0.003	1.131	−5.495	−4.365
	core length	137	1296	1433	−1.051	−1.290	−1.290	1.309	1.354	1.354	0.003	−0.094	−0.085	0.423	−121.678	−121.255
	core diameter	24	246	270	−0.211	−0.234	−0.234	0.200	0.241	0.241	−0.017	−0.015	−0.015	−0.413	−3.675	−4.088
	the number of rows of grain	54	428	482	−1.221	−1.455	−1.455	1.105	1.439	1.439	0.293	0.126	0.144	15.840	53.761	69.601
	the number of grains in a row	42	435	477	−2.755	−3.273	−3.273	2.523	2.662	2.662	−0.520	−1.048	−1.001	−21.844	−455.778	−477.622
	mass of grain from the cob	59	516	575	−13.690	−14.550	−14.550	12.970	13.830	13.830	1.344	−2.703	−2.288	79.270	−1394.760	−1315.490
	weight of one thousand grains	42	319	361	−26.780	−30.830	−30.830	28.510	33.290	33.290	0.946	−1.444	−1.166	39.730	−460.480	−420.750
	yield	59	516	575	−0.548	−0.582	−0.582	0.519	0.553	0.553	0.054	−0.108	−0.092	3.170	−55.796	−52.626

**Table 4 genes-13-00848-t004:** Characteristics and location of markers significantly related to the analyzed traits.

Marker	Marker Type	Chromosome	Marker Location	Associated with	Candidate Genes
17,300	DArT	Chr1	2.15 × 10^8^	cob diameter, the number of rows of grain, mass of grain from the cob, yield	40,523 bp at 5′ side: gdu1
					68,570 bp at 3′ side: receptor-like protein kinase isoform
18,852	DArT	Chr3	1.02 × 10^8^	cob diameter, the number of rows of grain, mass of grain from the cob, yield	25,4293 bp at 5′ side: low quality protein: peptidyl-prolyl cis-trans isomerase
					18,563 bp at 3′ side: uncharacterized protein loc103650335
1818	DArT	Chr8	1.5 × 10^8^	cob diameter, the number of rows of grain, mass of grain from the cob, yield	A marker that is anchored to the gene *cinnamoyl-CoA reductase 1*
16,474	DArT	Chr3	19,789,904	cob length, cob diameter, core length, core diameter	1270 bp at 5′ side: uncharacterized protein loc100382383 precursor
					4772 bp at 3′ side: uncharacterized protein loc100279241 precursor
14,506	DArT	Chr9	28,978,769	cob diameter, the number of rows of grain, mass of grain from the cob, yield	A marker that is anchored (WAT1-related protein At1g09380)
13,517	DArT	Chr9	1.31 × 10^8^	cob length, core length, the number of rows of grain, weight of one thousand grains	8016 bp at 5′ side: uncharacterized protein loc103639077 isoform 1
					450 bp at 3′ side: allene-oxide cyclase2
2317	DArT	Chr7	1.38 × 10^8^	cob diameter, the number of rows of grain, mass of grain from the cob, yield	A marker that is anchored (*eukaryotic translation initiation factor 3 subunit c*)
7950	DArT	Chr2	43,524,954	cob diameter, the number of rows of grain, mass of grain from the cob, yield	233,907 bp at 5′ side: actin binding protein precursor
					5461 bp at 3′ side: *mads-box transcription factor 27 isoform 2*
16,703	DArT	Chr2	1.68 × 10^8^	cob diameter, the number of rows of grain, mass of grain from the cob, yield	A marker that is anchored (uncharacterized protein loc100282883
17,490	DArT	Chr10	1.39 × 10^8^	cob length, cob diameter, core length, the number of rows of grain	91,320 bp at 5′ side: scarecrow-like protein 8
					6776 bp at 3′ side: uncharacterized protein loc100383502
17,843	DArT	Chr3	2.25 × 10^8^	cob length, the number of grains in a row, mass of grain from the cob, yield	1290 bp at 5′ side: uncharacterized protein loc100192921 isoform 1
					6791 bp at 3′ side: uncharacterized protein loc100276743
18,664	DArT	Chr5	2.11 × 10^8^	cob diameter, the number of rows of grain, mass of grain from the cob, yield	85,540 bp at 5′ side: uncharacterized protein loc100278506
					101,692 bp at 3′ side: *delta-12 fatty acid desaturasefad2 isoform 1*
3233	DArT	Chr3	2.1 × 10^8^	cob diameter, the number of rows of grain, mass of grain from the cob, yield	A marker that is anchored *RNA polymerase II transcriptional coactivator KELP*
4205	DArT	Chr5	2.26 × 10^8^	cob diameter, the number of rows of grain, mass of grain from the cob, yield	42,608 bp at 5′ side: uncharacterized protein loc118472127
					46,806 bp at 3′ side: *callose synthase*
11,657	DArT	Chr5	2.22 × 10^8^	cob diameter, core diameter, mass of grain from the cob, yield	A marker that is anchored *aspartate aminotransferase*
12,812	DArT	Chr1	15,198,950	cob length, core length, the number of rows of grain, weight of one thousand grains	A marker that is anchored *sucrose transporter 1*

## Data Availability

The datasets generated during and/or analyzed during the current study are available from the corresponding author on reasonable request.

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
