# Peer review of "The Use of DArTseq Technology to Identify New SNP and SilicoDArT Markers Related to the Yield-Related Traits Components in Maize"

_genes, 2022, doi:10.3390/genes13050848_

Round 1

Reviewer 1 Report

Dear authors,

your work contains very interesting research results and they are presented very comprehensively in the work. So I recommend your work.

Author Response

Dear Reviewer,

Thank you very much for your positive feedback and for checking our manuscript and recomendation

Reviewer 2 Report

The manuscript needs extensive revision for language and grammar. Some of the sentences are incomplete and they do not make sense, so it is sometimes hard to follow and understand the text. So, it is highly recommended that some native English speaker should be included in checking the language in the manuscript. Also, the authors need to pay more attention to the use of adequate expressions and technical terms, i.e. instead of “features of the yield structure” it should be yield components, etc. The whole manuscript should be better structured. It is evident that results of the research are interesting and useful for scientific community, but they have to be much better presented.

Figures/tables could be more appropriate because they are not easy to follow and understand.

Only the 35,7% of total of 42 references is current (published within last five years), so it is suggested that some more recent references in the field of research should be added. The references include only few self-citations.

Lines 80-85:  It would be better to exclude this sentence. By my opinion, it does not fit well with the rest of the text, and the application of listed methods is well explained in the paragraph that follows.

Constant repeating the enumeration of the examined traits throughout the text of the manuscript it is not necessary.

Lines 120-129: Results could be presented as a table to be more understandable

Lines 132-151: By my opinion, the whole segment should be re-written and presented more clearly

Lines 171-172: Based on genetic distance or on genetic similarity?

Line 175: “These lines were similar to the other genotypes in 41%” Which are “other genotypes”, it is not clear.

Line 516: Ding, J.; Ali, F.; Chen, G.; et al. Genome-wide association mapping reveals novel sources of resistance to northern corn 516 leaf blight in maize. BMC Plant Biol 201,. 15, 206. -Please, check the year in the reference

Round 2

Reviewer 2 Report

Authors responded and correctly answered to many of my comments, but there are still some important thing I want to adress.

The English is improved, but I think it should be further upgraded. For example, adequate term for yield and yield components is trait, not feature. Also, the term replication is more appropriate than repetitions. In general terms location is to position as locality is to village/town/area. Besides, since localities refer to predefined areas, they usually have names. Thus, the more adequate term is location than locality. And so on...These suggestions are aimed to improve the quality of scientific paper and the journal itself and I think it is neccessary to adopt them.

In the abstract there is a sentence: Genetic distance between the analyzed lines was estimated on the basis of which. – This sentence is uncomplete and there are several more sentences in the text which are the same. That is why I kindly asked for English to be carefully checked throughout the whole document and that is still my suggestion.

Lines 132-151: By my opinion, the whole segment should be re-written and presented more clearly.

I strongly recommend this segment to be presented more clearly, maybe only the specific correlations. It is not necessary to comment each one because they are already presented in the table.
